# Effect of Square Dance Interventions on Physical and Mental Health among Chinese Older Adults: A Systematic Review

**DOI:** 10.3390/ijerph19106181

**Published:** 2022-05-19

**Authors:** Kai-ling Ou, Ming Yu Claudia Wong, Pak Kwong Chung, Kei Yee Katie Chui

**Affiliations:** Department of Sport, Physical Education and Health, Hong Kong Baptist University, Hong Kong, China; 21482268@life.hkbu.edu.hk (K.-l.O.); pkchung@hkbu.edu.hk (P.K.C.); chuikatie@hkbu.edu.hk (K.Y.K.C.)

**Keywords:** older adults, Chinese, square dance, physical activity, systematic review

## Abstract

(1) Background: Square dancing is an emerging form of aerobic exercise in China, especially among middle-aged and older people. The benefits of square dancing have been investigated and promoted in recent years through research and interventions. Interventions have been conducted to promote the participants’ reactionary participation in physical activity, social and family cohesion, and other psychological benefits. Therefore, square dancing has been promoted as a major factor in China’s increase in physical activity prevalence. (2) Methods: A systematic review was used to identify studies that have indicated the effect of square dancing on the physical and mental health among Chinese older adults. (3) Results: Twenty-four studies examining the effects of square dancing on older Chinese adults were extracted. These studies were not found in English databases. The quality of the retrieved studies had a moderate-to-high risk of bias. Square dancing interventions were shown to result in effective mental, physical, and cognitive improvements in the systematic synthesis. (4) Conclusions: This study examined the effects of square dancing in China over the past 10 years on the physical and mental health of older adults. Based on the results of this study, recommendations can be made for future square dance interventions for older adults such as male-oriented, mixed-gender, or intergenerational programs.

## 1. Introduction

As a worldwide problem, the difficulties and challenges in the social security and public services brought about by the aging population have been widely studied. China, which is in a stage of rapid aging, has the largest older adult population worldwide [1]. In 2019, there were approximately 176 million people over 65 years of age, accounting for 12.6% of the total population [2]. In addition, adults aged over 65 years are expected to exceed 26.1% of the total population by 2050. Moreover, the trend of an aging population in Hong Kong is even more serious, where almost 31% of the population will be aged over 65 years by 2036 [3]. Therefore, as people live longer, it is essential for policymakers to focus on making older adults more active, improving their quality of life, and promoting their physical and mental health.

Physical activity has been a determinant of successful aging for decades [4]. The World Health Organization [5] recommends that at least 150 min of moderate-to-vigorous physical activity per week is beneficial for the health of older adults. Some evidence has illustrated that almost five million people died due to physical inactivity in 2008 [6], and “physical inactivity” was identified as the fourth-largest risk factor for non-communicable diseases in 2009. Additionally, studies have demonstrated that older adults with insufficient physical activity are more likely to suffer from physical and psychological deterioration, leading to higher morbidity and mortality due to a decrease in physical parameters such as strength, balance, and endurance; cardiovascular disease; diabetes; certain types of cancer; and mental health problems [7].

Furthermore, compared with young people, older people are more likely to experience negative life events such as bereavement [8], poor living conditions [9], loss of socioeconomic status after retirement [10], and fewer social networks [11]. All of these adverse factors may contribute to poor mental health such as depression, loneliness, and rumination [12,13], which in turn increases the risk of suicide, the utilization of medical services, and social costs [14]. Therefore, paying attention to the physical and mental health of older adults is key to promoting the implementation of active and successful aging.

Physical activity has been recommended as a preventive and therapeutic tool to promote the physical and mental health of older adults [15]. In addition, physical activity promotion programs used to improve physical and mental health have been extensively developed in recent years, and the meta-analysis results of randomized controlled trials (RCTs) are generally positive and effective [16,17,18]. Therefore, the potential of interventions to promote physical activity should be further explored. Compared to other kinds of traditional Chinese physical activity such as Kung Fu, Baduanjin, Qigong, Tai Chi Ball, and Tai Chi Quan [19], square dancing is an emerging form of aerobic exercise in China, which has not yet been involved in Chinese traditional exercise-related systematic reviews. Research and interventions have been conducted in recent years to investigate and promote the benefits of square dancing in cultivating people’s reactional physical activity participation, social and family cohesion, and other psychological benefits [20]. Square dancing has been indicated as an exercise favored by middle-aged and older people [21] and is not restricted by the environment. Hence, square dancing has been promoted as a key factor in the increase in China’s physical activity prevalence [22]. Recent studies have demonstrated the benefits of square dancing in terms of cognitive, mental, and physical health. Research has shown that older adult women with mild mental disorders experienced a reduction in depressive symptoms and better mental health quality of life after undergoing an 18-week square dancing intervention [19,23]. Research indicated that square dancing could significantly improve the participants’ family cohesion and subjective well-being. Moreover, square dancing has been shown to significantly reduce the risk of coronary heart disease in older adults [24] and increase the lower limb muscle strength and single-leg balance [25].

Therefore, the current study aimed to systematically summarize the impact of square dance over the past 10 years on the physical and mental health among older adults in China. The results of this study are expected to summarize the relationship between square dancing and the overall well-being of Chinese older adults as well as documenting the potential effect of square dance intervention on the physical and mental health of Chinese older adults including physical fitness, subjective well-being, and cognitive function. Based on the results, it will also provide a basis for information and recommendations for future square dance interventions for older adults.

## 2. Materials and Methods

### 2.1. Protocol and Registration

A systematic review was conducted based on the Preferred Reporting Items for Systematic Reviews and Meta-Analyses (PRISMA) guidelines [26]. The systematic review was registered (Register ID: CRD42022313728). We identified studies that have indicated the effect of square dancing on the physical and mental health among Chinese older adults.

#### 2.1.1. Eligibility Criteria and Database Search Process

Relevant studies published from 2006 onward were identified. Only intervention studies that indicated Chinese older adults aged 55 years or above were included in the data extraction. The keywords applied to the search were “Physical Activity OR Exercise”, “Older Adults OR Elderly OR Ageing People OR Older People OR Older Subject OR Aging OR Ageing OR Aged” AND “China” AND “Mental Health” AND “Physical Health” AND “Square Dance”. Based on the retrieved publication list from the scoping review, papers that indicated square dance, aerobic dance, or dance-related interventions were extracted and put forward for full-text analysis. A comprehensive search approach was used, which indicated both the published research from bibliographic databases and unpublished studies from the grey literature. The following bibliographic databases were searched: JSTOR, SCOPUS, SPORTDiscus, Web of Science, and the China Academic Journals Full-Text Database. In addition, reference lists of the highlighted systematic reviews were searched and used as data sources.

#### 2.1.2. Studies Selection and Data Collection Management

Based on the eligibility criteria, eligible titles and abstracts were identified, screened, and selected from a list of retrieved publications, and then involved in full-text screening. Mendeley reference management software was used to store and organize all relevant studies such as the screening and deletion of duplication records. Any limitations and criticisms arising from these studies were further examined using the proper risk of bias assessment tools.

#### 2.1.3. Methodological Quality Assessment

For the quality assessments, two reviewers independently took part in the assessment process, and consensus was reached when any criticism was raised. Appropriate quality assessment tools were assigned according to the research method. The Modified Cochrane Risk of Bias tool from the National Institutes of Health was used to assess the quality of the RCTs [27], as it involves the assessment of selection bias, performance bias, detection bias, attrition bias, reporting bias, and other important concerns that should be reported in the study. The Joanna Briggs Institute (JBI) Critical Appraisal Checklist [28] for Cohort Studies was used for the cohort studies, which includes the assessment of the reliability of the exposure of the participants, the distribution bias between the exposed and unexposed groups, reporting bias, appropriateness of the exposure and tests as well as the appropriateness of the statistical analysis. The quality of the studies was categorized as low (having a low-risk bias indicator of 70% or above), medium (50–69% of low-risk bias indicators), or high (having 49% or less low bias indicator) risk of bias.

#### 2.1.4. Data Extraction and Synthesis

Based on the purpose of indicating the effect of square dancing on the older adults’ physical and mental health, the effect outcomes of square dancing on any indicators of physical and mental health stated in the study were extracted. Moreover, the content description and duration of the square dancing intervention were extracted. Additionally, the physical and mental health measurements adopted in the studies as well as the demographic information (i.e., settings, age, and gender) of the studies were extracted. Data and information were synthesized using narrative analysis and a narrative summary approach [29].

## 3. Results

### 3.1. Demographic Information

A total of 24 studies that investigated the effects of square dancing on older Chinese adults were extracted. No studies were retrieved from English databases and were only from Chinese databases. All studies were based in China, with most of them having a higher proportion of female older adults. Figure 1 shows the PRISMA flow diagram of the study identification process.

### 3.2. Quality Assessment

The quality of the studies varied. Among the 12 randomized controlled trial studies, six were rated as having a high risk of bias, mainly due to the unclear statement of the participants’ group randomization process as well as the level of blinding throughout the intervention duration. The rest of the studies were indicated as having a low risk of bias, with group randomization and no selective reporting of results; however, some concerns of bias still existed in the blinding operations. Among the 12 cohort studies, four demonstrated a low risk of bias, with 73% of low-risk indicators; three studies demonstrated a low-to-medium risk of bias, with 64% of low-risk indicators; and five studies demonstrated a comparatively higher risk of bias, with 53% of low-risk indicators.

### 3.3. Intervention and Measurement

Among the retrieved publications, the square dance RCT studies’ intervention duration ranged from 12 to 60 weeks with an intervention frequency of twice a week for 50–60 min per session. For the cohort studies, the intervention duration tended to be shorter, between 12 and 24 weeks, with a higher frequency of three–five sessions per week lasting 30–90 min per session.

The intervention design of the square dance was not clearly stated in any of the retrieved studies. Only a few studies stated the design of the square dancing program, which involved dancing patterns of waltz, Viennese, tango, and rumba [30,31] as well as the basic forward, backward step, side slide, pony jump, and lunge [31,32].

The measurements used in the studies were inconsistent. Studies used the P300 electroencephalography measurement, the Scale of Elderly Cognitive Function [33], the Mini-Mental State Examination [34], the Montreal Cognitive Assessment Scale [35], and event-related potential brain response measure [36] for the cognitive function and brain response ability of older adults. The methods to measure the older adults’ well-being were diverse including the Physical Self-Esteem Scale, General Well-Being Scale [37], Bf-S Mood measurement [38], General Self-Efficacy and Self-Esteem [39], Gerotranscendence Scale for overall positive active aging [40], and SF-36 for quality of life [41]. The measures adopted to assess the older adults’ physical fitness and health tended to be similar, which included the measurement of body composition (i.e., weight, BMI, and fat-free mass), blood pressure, the handgrip strength test, and muscle strength tests (i.e., sit-up test, squad test, and all other kinds of fitness tests involved in the Senior Fitness Test Manual) [42]. Moreover, studies included different kinds of balance and stability tests such as the 8-foot up and go test, standing on one foot with eyes closed and/or open, and the swing feet test. Additionally, a few studies involved the vital capacity and VO^2^MAX tests to accurately measure the lung capacity of the participants. Table 1 showed the summary of the intervention studies. Table 2 showed summary of the cohort studies. Table 3 and Table 4 showed the quality assessment of intervention and cohort studies. 

### 3.4. Square Dance and Mental and Cognitive Health

Dance has a significant positive influence on the cognitive function of older adults [33], especially on their memory, visuospatial skills, executive function, and attention [31]. A study using a task-switching task has also shown a significant effect of square dance on older adults’ reaction time and performance accuracy [33]. Square dancing has also been shown to improve the overall mental well-being including self-esteem, positive aging, and quality of life of older adults [37,40].

### 3.5. Square Dance and Physical Health

The effect of square dancing on the physical fitness and health of older adults is cogent. Research has shown that dance is related to a considerably decreased risk of metabolic syndrome, reduced body fat and abdominal sedum thickness [38], improved cardiorespiratory function, reduced bone loss and osteoporosis, and lower blood pressure [43,44,45]. Square dancing demonstrated a strong influence on the older adults’ physical fitness, especially on static balance, lower limb muscle strength [30,46,47,48,49], and core strength [50], the effects of which are similar to those of Tai Chi.

**Table 1 ijerph-19-06181-t001:** Summary of the intervention studies.

Author and Year	Title	Location	N	Age	Sex (%Female)	Research Purpose	Study Design (Method)	Outcome Variables	Intervention Duration	Intervention Frequency/Intensity
Chen (2015) [42]	An experimental study on improving health fitness with square dancing for elder women	China, Bengbu	179	60.8	100%	To investigate the changes of health fitness for female older adults after square dance training.	RCT	Health fitness levels	6 months	4 times/week; 90 min/session
Chen (2014) [33]	A study of the effects of a square dance exercise intervention on cognitive function in older adults	China	125	66.73	48%	To explore the influence of the square dance movement on the cognitive function of older adults.	Pretest-post-test design	Cognitive function	12 months	4 to 5 times/week; 30 to 60 min/session,
Guo (2016) [34]	Efficacy of aerobic exercise training in the treatment of mild cognitive dysfunction in the elderly	Tangshan, China	47	65.8	63.80%	To explore the impact of aerobic exercise intervention on patients’ cognitive function among older patients with cognitive dysfunction in urban communities.	RCT	Cognitive function	3 months	3 times /week; 60 min/session
Li et al. (2012) [51]	Study on the effect of different exercise modalities on the state of mind of the elderly	Hengyang, China	120	60	N/A	To explore the influence of different exercise methods (including square dance) on the mood of older adults and provide references for them to choose effective exercise methods and promote mental health.	RCT	Mood states	6 months	5 times/week, 45–60 min/session (exercise intensity was maintained within (170-age) based on the heart rate)
Li (2020) [31]	An experimental study on the effect of fitness ballroom dancing on the cognitive ability of the elderly	Shijiazhuang, China	60	62.65	83.30%	To explore the effect of fitness dance on the cognitive ability of older adults.	RCT	Cognitive ability	6 months	3 times/week; 90 min/session
Li (2017) [37]	The effects of square dance exercise on older adults’ physical self-esteem and well-being	Pingdingshan, China	60	61.8	50%	To explore the influence of square dance on physical self-esteem and well-being of older adults.	Pretest-post-test design	Body self-esteem and well-being	3 months	3 times/week, 40 or more mins/session
Li (2015) [50]	A Comparative Study of the Effects of Northeast Dayang Opera and Taijiquan on the Core Strength of Older Adults	China, Jilin	90	62	50%	To analyze the influence of square dance on the core strength of older adults.	RCT	Functional fitness	8 weeks	6–7 times/week; 50–70 min/session
Luo et al. (2012) [41]	A study of the effects of different forms of exercise on the physical and mental health of the elderly	China	120	63.3	not mentioned	To explore the impact of different sports (including square dance) on the quality of life of older adults to provide a reference of effective exercise for them.	RCT	Physical and mental health	6 months	5 times/week; 45–60 min/session
Ma (2016) [30]	A study on the intervention model of aerobic dance and its effect on subjective well-being of middle-aged and elderly people	Shijiazhuang, China	96	60	74.00%	To explore the intervention mode of aerobics dance and its effect on older adults’ subjective well-being.	RCT	Subjective well-being	3 months	4 times/week; 45–60 min/session
Xie et al. (2012) [40]	A study of the effects of 3 exercise modalities on the elderly beyond aging	Hengyang, China	120	63.89	N/A	To explore the impact of three physical exercise programs (including square dance) on gerotranscendence of older adults and provide information for them to select effective exercise.	RCT	Gerotranscendence	6 months	5 times/week, 45–60 min/session (The exercise intensity was maintained within (170-age) based on the heart rate)
Xue (2017) [49]	A study on the intervention of square dance on physical balance ability of the elderly	China, Hebei	80	65	48.75%	To investigate the effect of square dance exercise, as an intervention tool, on the balance ability of older adults and provide a theoretical basis for better promotion of square dance to improve the balance ability of older adults.	RCT	Balance function	3 months	7 times/week; 75 min/session

**Table 2 ijerph-19-06181-t002:** Summary of the cohort studies.

Author and Year	Title	Location	N	Age	Sex (%Female)	Research Purpose	Study Design (Method)	Outcome Variables	Intervention Duration	Intervention-on Frequency/ Intensity
Fu (2011) [30]	Effects of physical dance on lower limb muscle strength and bone mineral density in older men	China, Shanghai	45	68.4	0%	To research and discuss the effects of sport dance on lower limb strength and bone mineral density (BMD) in the older male adults and present experimental and theoretical evidence for exercise prevention on lower limb exercise function degeneration as well as exercise prevention and treatment of senile osteoporosis.	Cohort study	Functional fitness	N/A	N/A
Li (2015) [36]	Effects of tai chi and square dance exercise on auditory event-related potentials P300 in the elderly	Zhengzhou, China	30	63	N/A	To study and compare the differences in the effects of square dance and Tai chi on auditory event-related potentials and the effects of different forms of exercise on human cognitive ability.	Cohort study	Cognitive function	>3 years	N/A
Liang (2016) [48]	A study on the effect of long-term different exercises on the static balance ability of the elderly	China, Shanghai	173	65	67%	To compare the differences in effect of 24 forms of simplified taijiquan, Yi Jin Jing and square dance on the indicators of static balance.	Cohort study	Balance ability	N/A	2–3 times/week; 90 min/session
Liu (2009) [32]	Comparative analysis of participation in taijiquan, ballroom dancing, and walking on static balance ability of older women	China, Hebei	172	66.51 Non exercise group: 67.35 ± 6.58	100%	To compare the differences between the static balance ability of older women who regularly participated in tai chi, ballroom dancing, and brisk walking with those who do not.	Cohort study	Balance ability	>1 year	3 times/week; 60 min/session
Liu et al. (2016) [35]	Wang, R.; Effects of different exercise programs on the cognitive abilities of older adults	Shanghai, China	262	63.5	N/A	To study the effects of different sports (including square dance) on cognitive ability of senior people.	Cohort study	Cognitive ability	>3 years	5 times/week; 60 min/session
Luan (2016) [46]	A study of the effects of different forms of fitness on the physical health of the elderly	China, Harbin	200	65	50%	To analyze the influence of four different exercises (including square dance) on the physical health of older adults and provide the basis for further study of older adults’ fitness theory.	Cohort study	Functional fitness	N/A	N/A
Sun & Wang (2020) [38]	The effects of square dance exercise on physical health and psychoemotional well-being of older adults.	China	80	60.87	70%	To study the influence of square dancing on the physical health and mental mood of older adults.	Cohort study	Functional fitness	3 months	3 times/week; 60 min/session
Sun (2017) [43]	A comparative study of the effects of different exercise programs on the health fitness of older adults	China, Nanning	N/A	65	N/A	To explore the effects of various sports (including square dance) on human health fitness.	Cohort study	Functional fitness	N/A	N/A
Wang (2009) [44]	A differential study on the effects of different types of leisure activities on physical function of older women	China, Chengdu	271	69	100%	To explore the effects of different types of leisure activities (including square dance) on the physical function of older adults.	Cohort study	Functional fitness	N/A	N/A
Wang (2014) [45]	A study on the effects of tai chi and fitness dance on the physical health of middle-aged and elderly women	China, Dalian	30	60.3	100%	To analyze the effects of different types of aerobic fitness exercises on the physical health of older adult women who usually have long-term regular adherence to tai chi or fitness dance exercises and those who do not have regular exercise habits.	Cohort study	Functional fitness	N/A	N/A
Zhou (2017) [46]	The effects of square dance and fitness walking exercise on functional fitness in older adults	China, Nanjing	207	64.8	77%	To investigate the effects of square dance and fitness walking on functional fitness of older adults.	Cohort study	Functional fitness	>1 year	3 times/week; 60 min/session

**Table 3 ijerph-19-06181-t003:** The Quality Assessment of Intervention Studies.

Author and Year	Random Sequence Generation	Allocation Concealment	Selective Reporting	Other Sources of Bias	Blinding (Participants and Personnel)	Blinding (Outcome Assessment)	Incomplete Outcome Data	Overall Quality
Chen (2015) [42]	Low risk of bias	Low risk of bias	Low risk of bias	Low risk of bias	Unclear	Unclear	Unclear	Low risk of bias
Chen (2014) [33]	Low risk of bias	Unclear	Low risk of bias	Low risk of bias	Unclear	Unclear	Unclear	High risk of bias
Guo (2016) [34]	Low risk of bias	Low risk of bias	Low risk of bias	Low risk of bias	Unclear	Unclear	Low risk of bias	Low risk of bias
Li et al. (2012) [51]	Low risk of bias	Unclear	Low risk of bias	Low risk of bias	Unclear	Unclear	Low risk of bias	Low risk of bias
Li (2020) [31]	Low risk of bias	Low risk of bias	Low risk of bias	Low risk of bias	Unclear	Low risk of bias	Low risk of bias	Low risk of bias
Li (2017) [37]	Low risk of bias	Unclear	Unclear	Unclear	Unclear	Unclear	Low risk of bias	High risk of bias
Li (2015) [50]	High risk of bias	Unclear	Low risk of bias	Low risk of bias	Unclear	Unclear	Unclear	High risk of bias
Luo et al. (2012) [41]	Low risk of bias	Low risk of bias	Low risk of bias	Low risk of bias	Unclear	Unclear	Low risk of bias	Low risk of bias
Ma (2016) [30]	High risk of bias	Unclear	Unclear	Unclear	Unclear	Unclear	Unclear	High risk of bias
Xie et al. (2012) [40]	High risk of bias	Unclear	Unclear	Unclear	Unclear	Unclear	Low risk of bias	High risk of bias
Xue (2017) [49]	Low risk of bias	Low risk of bias	Low risk of bias	Low risk of bias	Unclear	Unclear	Unclear	Low risk of bias

**Table 4 ijerph-19-06181-t004:** The Quality Assessment of Cohort Studies.

Author and Year	1. Were the Two Groups Similar and Recruited from the Same Population?	2. Were the Exposures Measured Similarly to Assign People to both Exposed and Unexposed Groups?	3. Was the Exposure Measured in a Valid and Reliable Way?	4. Were Confounding Factors Identified?	5. Were Strategies to Deal with Confounding Factors Stated?	6. Were the Groups/Participants Free of the Outcome at the Start of the Study (or at the Moment of Exposure)?	7. Were the Outcomes Measured in a Valid and Reliable Way?	8. Was the Follow up Time Reported and Sufficient to Be Long Enough for Outcomes to Occur?	9. Was Follow Up Complete, and if Not, Were the Reasons to Loss to Follow Up Described and Explored?	10. Were Strategies to Address Incomplete Follow Up Utilized?	11. Was Appropriate Statistical Analysis Used?	Risk of Bias %
Fu (2011) [30]	Yes	Yes	Yes	Yes	Yes	Yes	Yes	No	No	No	Yes	63%
Li (2015) [36]	Yes	Yes	Yes	Yes	No	Yes	Yes	No	No	No	Yes	63%
Liang (2016) [48]	Yes	Yes	Yes	Yes	Yes	Yes	Yes	No	No	No	Yes	73%
Liu (2009) [32]	Yes	Yes	Yes	Yes	Yes	Yes	Yes	No	No	No	Yes	73%
Liu et al. (2016) [35]	Yes	Yes	Yes	Yes	Yes	Yes	Yes	No	No	No	Yes	73%
Luan (2016) [46]	Yes	Yes	Yes	No	No	Yes	Yes	No	No	No	Yes	54%
Sun & Wang (2020) [38]	Yes	Yes	Unclear	Yes	Yes	Yes	Yes	No	No	No	Yes	63%
Sun (2017) [43]	Yes	Yes	Yes	No	No	Yes	Yes	No	No	No	Yes	54%
Wang (2009) [44]	Yes	Yes	Yes	Yes	No	No	Yes	No	No	No	Yes	54%
Wang (2014) [45]	Yes	Yes	Yes	No	No	Yes	Yes	No	No	No	Yes	54%
Zhou (2017) [46]	Yes	Yes	Yes	Yes	Yes	Yes	Yes	No	No	No	Yes	54%

## 4. Discussion

Overall, in line with a previous study, the majority of square dance participants are female [52] because women dance more for fitness, mood enhancement, trance, confidence, and escapism than men, whereas men dance mainly for intimacy motives [53]. In addition, in China, the gender-divided labor market has led middle-aged and older women to rediscover their sense of belonging and meaning in their lives by participating in this activity [54]. Future studies should consider collecting the perceptions of older male adult participants on square dancing [55] and designing male-oriented, mixed-gender, or intergenerational square dancing programs [56].

Given the various cognitive and mental scales used in review articles, square dancing has great potential in improving the different dimensions of psychological health among older adults. Similar to a previous systematic review, dancing has a positive effect on the older adults’ cognitive sense of adherence [57] and improves depression, loneliness, and negative emotions in older people with dementia [58]. Moreover, music, as the key factor in square dance, satisfies the older adults’ psychological needs [55], and group-based music and dance have the capacity to preserve psychological well-being in aging as well as combat social isolation [59,60]. Unlike Western dancing, Chinese square dancing is similar in form to congregational dance in that it is a collective exercise with elements of popular radio gymnastics that promote social interaction among participants [61]. Due to their early experiences in the collectivization era, older Chinese people have a strong sense of belonging to this rhythmically consistent and orderly form of dance. In addition, square dancing can address the psychological plight of empty-nest older adults by creating social interactions [54]. Previous anthropologists have also found that positive emotional states in synchronized dance groups lead to social closeness [62]. Future research should explore the psychological impact of musical rhymes, rhythms, and dance forms from different psychosocial cultural backgrounds on older adults.

Dance-based therapy has been proven to improve the balance, lower body strength, flexibility, and endurance, and prevent the fall risk of older adults [63]. Regarding the quality of intervention, the current studies lack standardized content of square-dancing interventions. As Chinese square dance is a fusion of various dance art forms and its content structure is broad, people can organize square dance in open areas whenever music is played. However, a lack of professional risk awareness, knowledge, and supervision can lead to varying degrees of injury in older adults [64,65]. Further study should emphasize the design and evaluation of the effectiveness of standardized square dance programs with different functions, bearing in mind the physical condition of older people. For example, developing evidence-based square dancing programs by selecting different physical functions, intensity, dance types, difficulties, and coaching type (trained or peer-led) or by developing technology sensors to trace the participants’ movements to reduce injury risk [56,66].

### 4.1. Strengths and Limitations

It is pertinent to note that square dancing is considered as a newly emerged physical activity, which has not yet been systematically studied from the perspective of sports. There was a lack of standardized content for the square-dance program or intervention. The sample size of this study was therefore relatively small, and it was only possible to retrieve data from Chinese databases. However, currently, square-dance interventions for older adults are beginning to attract community attention, with an increasing number of studies being published in international journals in English from 2020 onward [67,68,69]. This cultural-based collective movement is increasingly being viewed as complementary and alternative to the Western forms of PA [69]. Moreover, the current systematic review was able to summarize and document the potential effect of square dancing on the overall well-being of Chinese older adults including physical fitness ability, balance, physical health, subjective mental well-being, and cognitive function. As the review also summarized the elements of square-dancing intervention including the duration, dancing patterns, and poses, it is expected to provide an overview of what should be included in the square-dancing intervention for future intervention studies. Therefore, it is hoped that, in the future, more square-dance intervention programs can be developed not only for Chinese older adults, but also for older adults worldwide.

### 4.2. Future Implications

From the limitations above-mentioned, the reviewers can summarize that a potential effective square-dancing program should last for at least 12 weeks, three sessions per week, 60 min per session. The content of the square dancing should involve the basic dancing patterns, forward, backward step, side slide, and pony jump. The environmental setting of the square-dancing program is not restricted by venue, but outdoor parks might be affected by the weather. However, it will be an advantage to create standardized square-dancing pattern guidelines in order to reduce the variations in different studies. Moreover, the music components should be involved in the examination of square dancing. Music is expected to be an factor that affects the older adults’ physical performance, well-being, and physical activity enjoyment [69]. Additionally, gender differences, age differences, and physical ability differences should be taken into account in future interventions.

## 5. Conclusions

Physical activity has been recommended as a preventative and therapeutic tool to improve the physical and mental health of older adults. Square dancing is an emerging form of aerobic exercise in China, especially among middle-aged and older people. This study examined the potential effects of square dancing in China over the past 10 years on the physical and mental health of older adults, and revealed that square-dancing interventions were shown to be associated with effective mental, physical, and cognitive improvements in the systematic synthesis. There was, however, a significant gender difference among the participants. Furthermore, the exact effect of square dancing on the physical and mental health of older adults was not able to be indicated, unless a meta-analysis can be conducted in the future. Therefore, square dancing should be further promoted as a newly emerged sport in China as well as around the globe in promoting physical activity.

## Figures and Tables

**Figure 1 ijerph-19-06181-f001:**
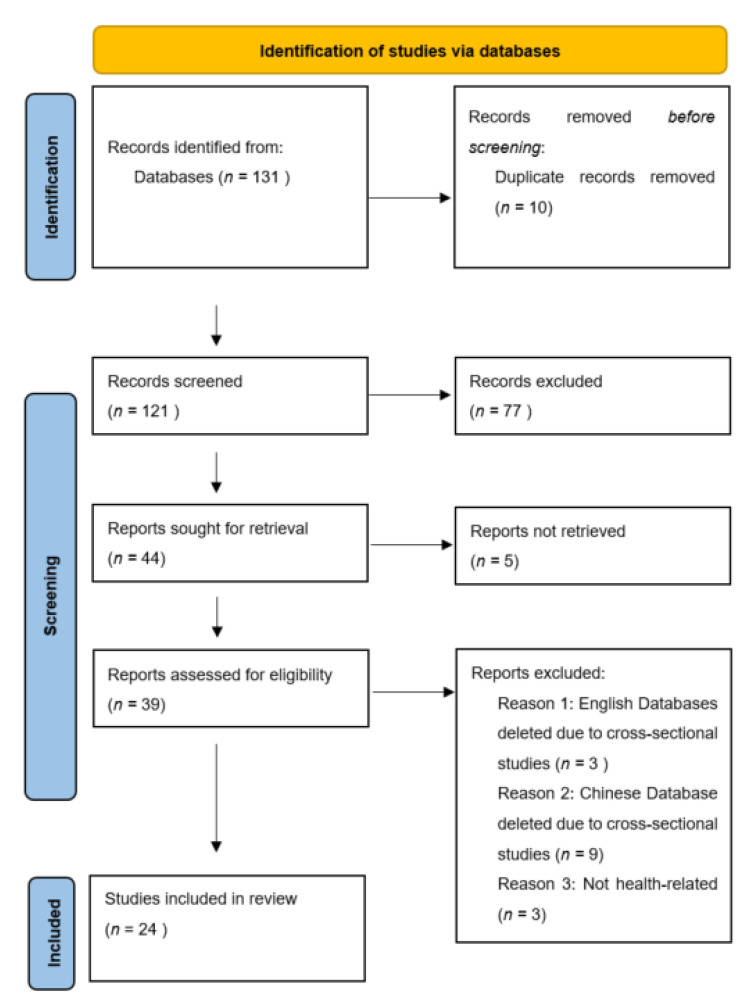
The PRISMA flow diagram.

## Data Availability

Not applicable.

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
