# Peer review of "Effect of Square Dance Interventions on Physical and Mental Health among Chinese Older Adults: A Systematic Review"

_ijerph, 2022, doi:10.3390/ijerph19106181_

Round 1

Reviewer 1 Report

The study is innovative and addresses important information on the physical and mental health among Chinese older adults. I recommend its publication after minor changes .

1.The general objective and specific objectives should appear at the end of the introduction. The objective should be clearly written, referring to the population, the intervention, the comparison and the results (PICO strategy)

2.The sample is very small for this type of study that should pay careful attention to its inference results, and should be limited in the article.

3. This article draws on physical and mental health among Chinese older adults.  that should pay careful attention to its inference results, and should be limited in the article. In the last paragraph of the conclusion, I recommend you to include the limitations of the study.

4.The form should refer to the MDPI Style Guide.

Author Response

We would like to thank the reviewer for their deliberate reviews of the research article. The raised considerable concerns are very helpful for improving the article. We agree with almost all their comments that we have revised the article and responded to the comments accordingly.

The detailed responses to each of the reviewers’ comments will be stated below. We clearly stated the revised parts with the particular paragraph shown, as well as indicating the page for referring to the paper; or if we have slightly countered with some of the points, we stated the reason with supporting literature. We hope that the reviewers will find our responses persuasive and cogent, and we are willing to accept further suggestions that the reviewers may have.

  In the following, each response is targeting each reviewers’ comments, the comments are in italics with responses inserted after it.

Reviewer 1

The study is innovative and addresses important information on the physical and mental health among Chinese older adults. I recommend its publication after minor changes .

1.The general objective and specific objectives should appear at the end of the introduction. The objective should be clearly written, referring to the population, the intervention, the comparison and the results (PICO strategy)

Response: Thank you for your suggestions. In order to make the study objectives clear, elaboration was made at the end of the introduction with track changes.

2.The sample is very small for this type of study that should pay careful attention to its inference results, and should be limited in the article.

  1. This article draws on physical and mental health among Chinese older adults.  that should pay careful attention to its inference results, and should be limited in the article. In the last paragraph of the conclusion, I recommend you to include the limitations of the study.

Response: In cooperating the 2 comments, the respective limitations were stated in the discussion and conclusion session. The wordings of “effect”, “resulted in” were mostly modified.

4.The form should refer to the MDPI Style Guide.

Response: The respective references were corrected.

Reviewer 2 Report

In this study, the authors examined the effects of square dancing on physical and mental health in Chinese older adults. Square dancing interventions were shown to be effective mental, physical, and cognitive improvements.  
The place of origin of square dancing is Western countries, especially United States. In China, it is a newly emerged sports, and seems to be not so common.

・Please mentioned the previous reports in effects of square dancing on physical and mental health in Western countries in more detail.

・Please explain the superiority of square dancing to other sports, for example a kind of traditional Chinese shadow boxing.

Author Response

We would like to thank the reviewer for their deliberate reviews of the research article. The raised considerable concerns are very helpful for improving the article. We agree with almost all their comments that we have revised the article and responded to the comments accordingly.

The detailed responses to each of the reviewers’ comments will be stated below. We clearly stated the revised parts with the particular paragraph shown, as well as indicating the page for referring to the paper; or if we have slightly countered with some of the points, we stated the reason with supporting literature. We hope that the reviewers will find our responses persuasive and cogent, and we are willing to accept further suggestions that the reviewers may have.

  In the following, each response is targeting each reviewers’ comments, the comments are in italics with responses inserted after it.

Reviewer 2

In this study, the authors examined the effects of square dancing on physical and mental health in Chinese older adults. Square dancing interventions were shown to be effective mental, physical, and cognitive improvements.  
The place of origin of square dancing is Western countries, especially United States. In China, it is a newly emerged sports, and seems to be not so common.

・Please mentioned the previous reports in effects of square dancing on physical and mental health in Western countries in more detail.

Response: Thank you for raising this comment. While, the authors would like to explain that, up till the date of responding the reviews, as well as referring to what we have mentioned in the manuscript (at the end of the discussion session), we could not find any square dancing related research that were based in the western countries. We would like to point out that, there were increasing number of research papers regarding square dancing among Chinese older adults, have been published in English. Our discussion session is also aimed at stating that more English written research could further promote square dancing beyond the Chinese population.

・Please explain the superiority of square dancing to other sports, for example a kind of traditional Chinese shadow boxing.

Response: The superiority of square dancing, compared to other Chinese sports is, square dance is considered as a newly emerged sport which has not yet been deeply studied. Systematic reviews that studied Chinese traditional sports did not involve square dance as well. Also, Chinese shadow boxing was also named as Tai Chi Quan, which it is obvious that Tai Chi related exercises were highly examined in the field. The related statements and elaboration are stated in the manuscript.

Reviewer 3 Report

I thank the authors for presenting an interesting topic. However, I think some adjustments are necessary.

The introduction is solid and well written, just asking them to updete some references. For example, review the latest physical activity recommendations (Bull et al. World Healt Organization 2020 guidelines on physical activity and sedentary behaviour. British Journal of Sports Medicine. 2020; 54(24): 1451-62).

Check the edition the citations on lines 47 and 57.

In the methodology, please add the citation regarding the PRISMA version that was used (Page et al., 2020).

I suggest reviewing the terms (MeSH) for the search, for example for older people, I suggest: elderly OR older adults OR older people OR older subject OR aging OR ageing OR aged 

The results are mixed with the methodology, although they are intersting, I suggest separating the methodological aspects in paragraphs, for example: protocol and registration, eligibility criteria, information and database search process, studies selection and data collection process, methodological quality assessment, data synthesis, certainty of evidence. Subsequently, accurately describe the results of the review (the flowchart is part of the results)

In think that the discussioin could be developed further, I suggest adding the limitation ans strengths of the review, as well as practical implications.

I suggest checking:

Zubala, A.; MacGillivray, S.; Frost, H.; Kroll, T.; Skelton, D.A.; Gavine, A.; Gray, N.M.; Toma, M.; Morris, J. Promotion of physical activity interventions for community dwelling older adults: A systematic review of reviews. PloS one 2017, 12, e0180902. 

Bromley, S.J.; Drew, M.K.; Talpey, S.; McIntosh, A.S.; Finch, C.F. A systematic review of prospective epidemiological research into injury and illness in Olympic combat sport. British journal of sports medicine 2018, 52, 8-16.

Valdés-Badilla, P.; Ramirez-Campillo, R.; Herrera-Valenzuela, T.; Branco, B.H.M.; Guzmán-Muñoz, E.; Mendez-Rebolledo, G.; Concha-Cisternas, Y.; Hernandez-Martínez, J. Effectiveness of Olympic Combat Sports on Balance, Fall Risk or Falls in Older Adults: A Systematic Review. Biology 2022, 11, 74. 

Thomas, E.; Battaglia, G.; Patti, A.; Brusa, J.; Leonardi, V.; Palma, A.; Bellafiore, M. Physical activity programs for balance and fall prevention in elderly: A systematic review. Medicine 2019, 98, e16218. 

Neville, C.; Nguyen, H.; Ross, K.; Wingood, M.; Peterson, E.W.; DeWitt, J.E.; Moore, J.; King, M.J.; Atanelov, L.; White, J.; et al. Lower-limb factors associated with balance and falls in older adults: A systematic review and clinical synthesis. J. Am. Podiatr. Med. Assoc. 2020, 110, Article_4.

Zhong, D.; Xiao, Q.; Xiao, X.; Li, Y.; Ye, J.; Xia, L.; Zhang, C.; Li, J.; Zheng, H.; Jin, R. Tai Chi for improving balance and reducing falls: An overview of 14 systematic reviews. Ann. Phys. Rehabil. Med. 2020, 63, 505–517.

Author Response

We would like to thank the reviewer for their deliberate reviews of the research article. The raised considerable concerns are very helpful for improving the article. We agree with almost all their comments that we have revised the article and responded to the comments accordingly.

The detailed responses to each of the reviewers’ comments will be stated below. We clearly stated the revised parts with the particular paragraph shown, as well as indicating the page for referring to the paper; or if we have slightly countered with some of the points, we stated the reason with supporting literature. We hope that the reviewers will find our responses persuasive and cogent, and we are willing to accept further suggestions that the reviewers may have.

  In the following, each response is targeting each reviewers’ comments, the comments are in italics with responses inserted after it.

Reviewer 3

I thank the authors for presenting an interesting topic. However, I think some adjustments are necessary.

The introduction is solid and well written, just asking them to update some references. For example, review the latest physical activity recommendations (Bull et al. World Health Organization 2020 guidelines on physical activity and sedentary behaviour. British Journal of Sports Medicine. 2020; 54(24): 1451-62).

Check the edition the citations on lines 47 and 57.

In the methodology, please add the citation regarding the PRISMA version that was used (Page et al., 2020).

Response: The respective references were added and modified.

I suggest reviewing the terms (MeSH) for the search, for example for older people, I suggest: elderly OR older adults OR older people OR older subject OR aging OR ageing OR aged 

Response: The respective terms are modified.

The results are mixed with the methodology, although they are interesting, I suggest separating the methodological aspects in paragraphs, for example: protocol and registration, eligibility criteria, information and database search process, studies selection and data collection process, methodological quality assessment, data synthesis, certainty of evidence. Subsequently, accurately describe the results of the review (the flowchart is part of the results)

Response: Thank you for the suggestions. Sub-titles were added into the methodology part to clearly indicate the respective information.

In think that the discussion could be developed further, I suggest adding the limitation and strengths of the review, as well as practical implications.

Response: Thank you for your suggestions and references for our information. Taking those references as reference, the authors have elaborated the discussion session by adding the limitation and strengths, as well as Future implications sub-sessions.

Reviewer 4 Report

From the methodological point of view, the paper is very well written, the aims are clearly defined as well as the material and methods, results, discussion and conclusions. It also includes the limitations and study of the possible biases. However, I have  some doubts. The benefits of the exercise to increase the quality of life in the elderly are sufficiently studied. What element of difference does it include square it dances regarding the exercise in general?. Also,  the inclusion and exclusion criteria of the articles including should be revised.

To make bibliographical appointments in Chinese makes impossible their confirmation and relevancy in the article.

Why date number eleven is diferent from others? line 47

Author Response

We would like to thank the reviewer for their deliberate reviews of the research article. The raised considerable concerns are very helpful for improving the article. We agree with almost all their comments that we have revised the article and responded to the comments accordingly.

The detailed responses to each of the reviewers’ comments will be stated below. We clearly stated the revised parts with the particular paragraph shown, as well as indicating the page for referring to the paper; or if we have slightly countered with some of the points, we stated the reason with supporting literature. We hope that the reviewers will find our responses persuasive and cogent, and we are willing to accept further suggestions that the reviewers may have.

  In the following, each response is targeting each reviewers’ comments, the comments are in italics with responses inserted after it.

Reviewer 4

From the methodological point of view, the paper is very well written, the aims are clearly defined as well as the material and methods, results, discussion and conclusions. It also includes the limitations and study of the possible biases. However, I have  some doubts. The benefits of the exercise to increase the quality of life in the elderly are sufficiently studied. What element of difference does it include square it dances regarding the exercise in general?. Also,  the inclusion and exclusion criteria of the articles including should be revised.

Response: Thank you for your affirmation. While, in regards to well-documented outcomes of physical activity and exercises, the authors would like to point out that square dance is considered as a newly emerged sport in China, the studies on square dance are still considered as developing. Despite the effect of exercises on physical and mental health is well-documented in the field, the purpose of the review is to provide the field with a new form of physical activity, and to provide the community with an alternative of physical activity. Hence, beforehand, the effect of square dancing should be summarized and indicated, the characteristics of square dancing, such as the intervention duration, content and advantages, are also explained in the manuscript. The current review is also aimed at providing a basis for information and recommendations for future square dance interventions for older adults.

To make bibliographical appointments in Chinese makes impossible their confirmation and relevancy in the article.

Response: According to the reference guidelines of non-English written references, the titles are translated into English and stated in the square bracket following the original language of the title.

Why date number eleven is different from others? line 47

Response: The respective in-text citation is deleted.

Round 2

Reviewer 3 Report

I thank the authors for the new version of their manuscript, wich is much better. However, some minor adjustments need to be made:

  • Review the citation from Page et al. (2017), it should be 2020.
  • In line 97 you must change or x OR
  • In the methodology the must briefly describe the instruments used to assessment the methodological quality
  • Align reference 19

Author Response

Dear Reviewer, 

Thank you for the comments and suggestions that have been pointed out. 

The required information regarding the quality assessment tool has been inputted by stating the assessment components of the tools, and the required references and citation has been corrected. 

Thank you.

Reviewer 4 Report

Thank you very much for clarifying the doubts after the first reading of the paper. As I said in the first revision, from a methodological point of view, the paper is correct. I regret the error on my part of putting major revision when it was a minor revision. I apologise for that. 

Author Response

Dear Reviewer, 

Thank you for your affirmation. Hope you enjoy reading our review paper. 

Thank you.